# Replicating different roles of intent across moral domains

## Joseph Sweetman and George A. Newman

Department of Psychology, College of Life and Environmental Sciences, University of Exeter, Exeter EX4 4QG, UK

 JS, 0000-0002-2829-1503; GAN, 0000-0002-2322-108X

## Replication

psychology

morality, moral domains, intent, purity, harm, theory of mind

**Author for correspondence:**
Joseph Sweetman
e-mail: j.p.sweetman@exeter.ac.uk

Whether moral cognition is underpinned by distinct mental systems that process different domains of moral information (moral pluralism) is an important question for moral cognition research. The reduced importance of intent (intentional versus accidental action) when judging purity (e.g. incest), when compared with harm (e.g. poisoning), moral violations is, arguably, some of the strongest experimental evidence for distinct moral systems or 'foundations'. The experiment presented here is a replication attempt of these experimental findings. A pre-registered replication of Experiment 1B from the original article documenting this effect was conducted in a sample of $N = 400$ participants. Findings from this successful replication are discussed in terms of theoretical and methodological implications for approaches to moral cognition.

## 1. Introduction

Imagine siblings having consensual sex. This is probably difficult to picture. Now imagine somebody causing physical harm to their sibling. This is not such a difficult task, at least for those that have experienced sibling rivalry. An even easier task though is to assign a moral status to each of these actions. This is something that people do intuitively across an endless variety of novel actions [1]. Moral pluralism suggests that our moral judgements of such cases involving harm and incest are underpinned by separate, domain-specific moral modules[1] or systems. For example, moral foundations theory (MFT) posits that humans have distinct mental modules or 'foundations' that process moral information across a specific range of moral domains and their associated actions: harm (e.g. assault), purity (e.g. incest), fairness (e.g. cheating), loyalty (e.g. betrayal) and authority (e.g. disobedience) [2,3]. MFT claims that these modules each process information in a bespoke manner

---

[1]Nothing in the study presented here hangs on the acceptance of modularity. For our purposes, distinct moral systems could be entirely learned by domain-general learning processes. Rather, we are interested in the replicability of the evidence in support of separate moral systems, regardless of whether they reflect the innate structure of domain-specific learning modules.

that reflects the particular selection pressures responsible for sculpting the evolution of that mental module. For example, it posits that the harm system or foundation helps to meet the adaptive challenge of caring for vulnerable offspring across an extended period of time, a common challenge for all mammals. It does this by mapping perceptions of suffering (e.g. seeing someone being assaulted) to relevant moral cognitions (e.g. feeling that this is morally wrong) and actions (e.g. motivations to help and protect).

Critics accuse moral pluralism of lacking parsimony, conceptual clarity, and methodological rigour [4,5]. More generally, some evolutionary biologists have argued that while it is easy to give plausible adaptationist 'stories' for the evolution of human cognition, it is much harder, perhaps even impossible, to empirically test these as scientific hypotheses [6]. Regardless of whether we can answer how human moral cognition evolved, understanding whether it is accurately characterized by a single system or separate moral systems (moral pluralism) is an important question in its own right. Indeed, correctly delineating and decomposing the phenomenon under inquiry is one of the standard steps for developing theoretical explanation in the psychological and cognitive (neuro)sciences [7].

One way of proceeding to delineate and decompose cognitive phenomenon (e.g. memory) is to demonstrate that some experimental manipulation 'x' (e.g. typeface) causally impacts one (sub)type of the phenomenon '$y_1$' (e.g. implicit memory) differently from how it affects another (sub)type of the phenomenon '$y_2$' (e.g. explicit memory) [7]. This approach has been taken with regard to the role of intent across the moral domains of harm and purity, and provides, arguably, some of the strongest experimental evidence for moral pluralism's claim that our moral cognition is underpinned by separate moral modules or systems.

## 2. Different roles of intent across moral domains

Given intent plays a crucial role in moral judgements of harm (e.g. murder versus manslaughter), Young & Saxe [8] set out to examine whether this was also the case for purity violations (e.g. incest, eating taboo foods). On the basis of anthropological observations, they predicted that intent would not exculpate purity violations to the same extent as harm violations. They found just that: the exculpatory effect of innocent intentions was significantly reduced for purity compared to harm violations. For example, accidentally poisoning a cousin with an undisclosed peanut allergy was judged less morally wrong than accidentally committing incest with a long-lost sibling (Experiments 1–3). However, intentional harm was either judged the same as, or worse than, intentional purity violations. Put differently, the simple main effect of mental state (intentional versus accidental) was stronger for harm than purity violations. This is taken as evidence for the separate cognitive processes or 'signatures' underpinning moral judgements in these domains. Young & Saxe draw on, admittedly speculative, functional explanations to account for the different cognitive processes underpinning moral judgements of harm and purity violations. For example, they suggested that intent is important for harm violations as it helps us identify who will cooperate with us in the future. By contrast, such information is less important for purity violations where it is the act, rather than the intention, that it said to serve the function of signalling group membership and cohesion.

There has been further converging evidence for, or 'conceptual replication' of, the mental state × domain interaction [9,10]. However, pre-registered, independent, close or 'direct' replication of such a key result is important if we are to better understand the mechanisms underlying moral cognition. More specifically, having replicable phenomena to explain is a necessary prerequisite for the difficult task of discovering the mechanisms that explain such phenomena.

## 3. Replication of Experiment 1B

We aimed to independently replicate Young & Saxe's finding for the reduced exculpatory value of innocent intentions in purity (versus harm) violations. Although Young & Saxe demonstrated the targeted effect in four separate experiments (Experiments 1A, 1B, 2 and 3), we planned a close replication of Experiment 1B. We chose this particular experiment for two reasons. First, it was the only study which presented moral dilemmas in the third-person (e.g. Sam knows/has no idea). The remaining studies presented dilemmas in the second-person (e.g. you know/have no idea). This is potentially problematic as it has been argued, convincingly, that the use of second-person (versus third-person) dilemmas risks increasing the role of exogenous (external to core moral cognition) factors [11]. Therefore, it is important to replicate this effect employing dilemmas in the third-person. Second, this experiment had the smallest sample size of Experiments 1A ($N = 262$), 1B ($N = 80$),

2 ($N = 320$) and 3 ($N = 160$). Replication is important in this case as small samples are likely to overestimate effect sizes and lead to low reproducibility of results [12].

## 3.1. Method

This article received in-principle acceptance (IPA) at Royal Society Open Science on 6 June 2019. Following IPA, the accepted Stage 1 version of the manuscript was pre-registered on the OSF at https://osf.io/wjg63/. This pre-registration was performed prior to data collection and analysis. Furthermore, all materials, data and analysis scripts are openly available in the OSF online repository at https://osf.io/t7mzb/. Our pre-registered method employs the detailed materials supplied as supplementary material for the original article. We contacted the first author asking for their comments on our pre-registration plan and materials. They immediately responded that they were happy with the replication materials, but later clarified that mental state information was never presented to participants in bold, as could be inferred from the original supplementary material. Any differences from the originally reported study or our pre-registration are clearly explained in the following sections.

### 3.1.1. Participants and design

Four hundred participants (167 men, 227 women, 5 unspecified) were recruited through the online crowdsourcing platform Prolific (prolific.ac), ages ranged from 18 to 64 years ($M = 30.49$, s.d. $= 9.82$). The experiment took approximately 1 min to complete. Participants were, therefore, paid £0.09 to complete the study at an hourly rate of £5.40. Young & Saxe employed the MTurk platform to sample their participants. We chose to use Prolific as our crowdsourcing platform because of the ease of accessing existing credit to pay participants on this platform. Prolific samples have been shown to replicate known effects and to produce similar quality data as MTurk samples, and they have the added advantage of more naive, and less dishonest, participants compared to MTurk [13,14].

The experiment was a 2 (mental state: intentional versus accidental) × 2 (domain: harm versus purity) between-participants design. Based on the reported effect size estimate of $\eta_p^2 = 0.07$ for the mental state × domain interaction, only 175 participants were required to achieve 95% power in a 2 × 2-design with an α-level of 0.05. We assume that the true effect size may be smaller than reported. We thus aimed for a sample of 400 participants, which is five times the original sample size of 80. This would result in a power of 95% to detect an effect $\eta_p^2 = 0.032$, an effect less than half the size of that reported.

### 3.1.2. Materials and procedure

Participants made a moral judgement of the actions of an agent (Sam) in one of the four scenarios. The harm violation involved intentionally (or accidentally) poisoning a cousin, while the purity violation involved intentionally (or accidentally) engaging in incest with a long-lost sibling (see Materials at https://osf.io/t7mzb/). Participants judged the moral wrongness of the action described in the scenarios on a 7-point scale, anchored at (1) *not at all morally wrong* to (7) *very morally wrong*. The scenarios and response scale were taken directly from Experiment 1B of the supplementary material supplied in the original article. The experiment was conducted within the Qualtrics environment (contact the primary author for a copy of the materials in the Qualtrics environment).

### 3.1.3. Statistical analysis

The analyses followed that in the original paper. Specifically, we tested the mental state × domain interaction using a two-way ANOVA and planned comparisons. We supplemented this original approach with Bayesian hypothesis testing. Bayes factors have some advantages over traditional frequentist approaches (*p*-values) and are now becoming commonplace in psychological and cognitive (neuro)science research, especially when we are interested in a symmetrical measure of evidence for the alternative and null hypotheses [15]. As stated in the pre-registration, we excluded data from participants that completed the study in under 7 s (the minimum time needed to read and respond to the scenario and moral judgement question). We also excluded data from participants ($N = 17$) reporting that they had previously completed the same study (see pre-registration and supplementary robustness analyses for details at https://osf.io/t7mzb/). Finally, as stated in the pre-registration, we evaluated our replication results based on a number of approaches: vote-counting based on *p*-values

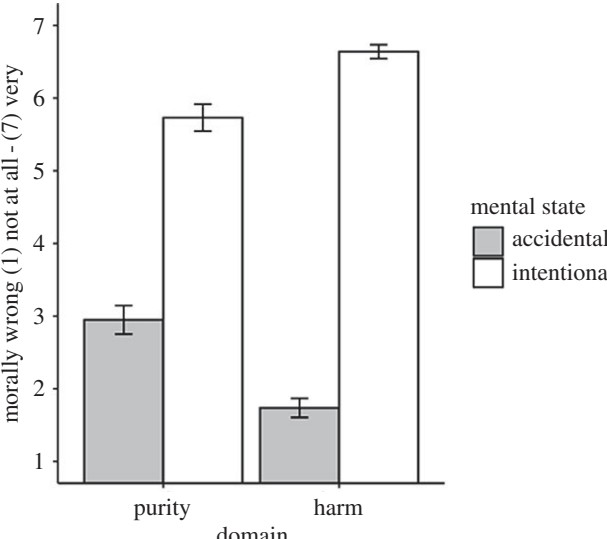

**Figure 1.** Moral judgements as a function of mental state (intentional versus accidental) and domain (harm versus purity). Error bars reflect 95% CIs.

and effect sizes, confidence intervals and the 'small telescopes' approach [16] and replication Bayes factors for ANOVA results [17].

# 4. Results

First, we report the results for the mental state × domain interaction and then summarize the success of the replication attempt from several perspectives. Analyses were completed using R. The data and scripts for all analyses are available at https://osf.io/t7mzb/.

## 4.1. Mental state × domain effect

The mental state × domain interaction was statistically significant, $F_{1, 379} = 45.50$, $p < 0.001$, $\eta_p^2 = 0.11$, 95% CI [0.06, 0.16]; $BF_{10} = 1.36 \times 10^8 \pm 7.3\%$ for the full model compared with the main effects-only model. As seen in figure 1, planned comparisons revealed that the exculpatory effect of innocent intentions (i.e. the simple main effect of mental state) was significant, with greater magnitude, for the harm violation ($t_{379} = -22.09$, $p < 0.001$; $BF_{10} = 6.47 \times 10^{66} \pm 2.9\%$ for the unconstrained model compared with the simple-effect-constrained model) compared to the purity violation ($t_{379} = -12.50$, $p < 0.001$; $BF_{10} = 2.78 \times 10^{27} \pm 15\%$ for the unconstrained model compared with the simple-effect-constrained model). Taken together, these findings suggest strong evidence for the exculpatory effect of innocent intentions being greater for harm compared to purity violations.

## 4.2. Evaluation of replication success

Considering the estimated effect sizes and their respective 95% CIs (figure 2), our replication has higher power and a consistent, but slightly larger, effect size estimate than the original study. All confidence intervals for $d$ exclude zero. The 'small telescopes' approach [16] suggested that the original study had 33% power to detect a 'small effect' of $d33\% = 0.5$. As seen in figure 2, the original study's confidence interval includes the 'small effect', while our replication does not. Thus, our replication is consistent with the notion that the studied effect is large enough to have been detectable with the original sample size.

By testing the hypothesis that the effect in a replication study is consistent with the original finding against the hypothesis that there is no effect, the replication Bayes factor provides relative evidence for, or against, a successful replication [17]. The replication Bayes factor for the mental state × domain effect was $BF_{r0} = 4.7 \times 10^8$, indicating strong evidence in favour of a consistent effect. Put differently, the model with a prior based on the original study's posterior is about $4.7 \times 10^8$ times more likely than the model with an effect size of $f^2 = 0$, assuming balanced groups in both studies. Taken together, these findings offer strong support for a successful replication of the original mental state × domain effect.

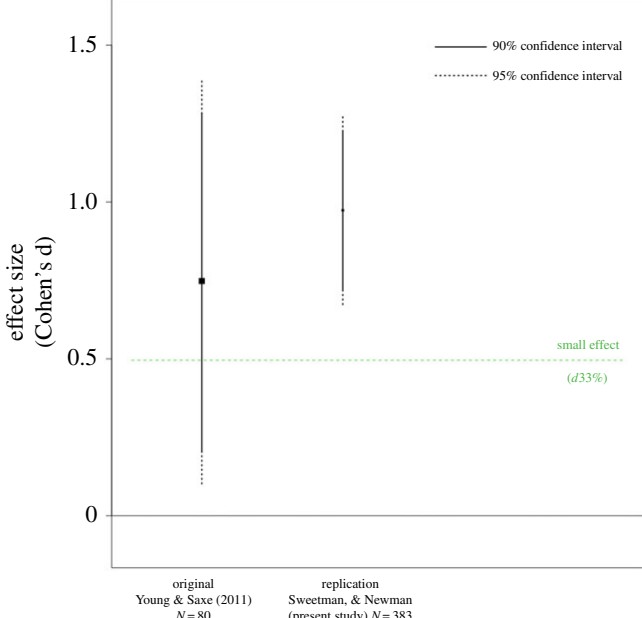

**Figure 2.** Results from Young & Saxe's [8] Experiment 1B and our replication (Sweetman & Newman, present study). Effect-size estimates and their confidence intervals are plotted. The dashed line indicates the effect size ($d33\% = 0.5$) that would give the original study, with a sample size of 20 per cell, 33% power.

## 5. Discussion

We find strong evidence for the mental state × domain interaction effect. Specifically, we successfully replicate the reduced exculpatory value of innocent intentions in purity (versus harm) violations reported in Experiment 1B of Young & Saxe's original article [8]. This provides the first pre-registered, independent, close replication of the mental state × domain interaction effect. Taken together with other converging fMRI and cross-cultural evidence [9,10], the mental state × domain interaction effect seems like a reliable and robust effect. Despite most studies of the effect employing second-person (versus third-person) dilemmas that risk increasing the role of exogenous factors [11], we, along with others [10], find evidence for the effect when dilemmas are presented in the third-person. This suggests that the effect cannot be explained by any increase in exogenous factors associated with the use of second-person moral dilemmas.

More broadly, our findings offer more support for the idea that our moral judgements of cases involving harm and purity are underpinned by separate, domain-specific moral modules or systems [2,3]. That said, simply because we replicate an effect does not mean that the theory said to explain the effect is correct [18]. There are other (than moral pluralism) explanations of the effect that need to be tested in future research. Harm and purity dilemmas employed in the literature differ in more than simply their moral domain. For instance, there is evidence that the purity dilemmas employed in the literature are 'weirder' and less severe than the corresponding harm dilemmas [5]. These unresolved confounds of content differences with moral domain must be systematically addressed in future research. Indeed, such differences may point towards explanations of the mental state × domain interaction effect that are more domain-general in nature. Huebner *et al.* [19] offer a similar perspective when considering the role of emotion in moral psychology. Specifically, the authors argued that emotion could impact moral cognition through its impact on attentional processes. Analogously, the weirdness of purity (versus harm) scenarios may compete with standard moral cognition for attentional resources. This could lead to a failure to integrate mental state information into the computations underlying moral cognition in these cases. This may explain the reduced exculpatory effect of innocent intentions in the purity (versus harm) domain without having to evoke separate, domain-specific moral systems. Future research is needed to test among domain-specific and more domain-general explanations of the effect.

An important limitation of the present replication attempt is that it is based on one sole set of stimuli for each domain—namely, poisoning a cousin (harm) and engaging in incest with a long-lost sibling (purity). The extent to which we can generalize to all harm and purity violations is uncertain. Stimuli

sampling issues are a general problem for social cognition research [20] and have been shown to limit the generalizability of classic effects in moral cognition research [21]. Experiments 1A, 2 and 3 in the original Young & Saxe study employed two sets of stimuli for harm and eight sets (four incest and four ingestion of taboo substances) for purity. However, it seems that these were not analysed in a way that would address stimuli sampling issues [20]. We can gain some confidence in the generalizability of the effect to physical harm, incest and ingestion of taboo substances from more recent research that has found the effect with item-wise analysis, employing 24 sets of physical harm and 24 sets of purity (12 incest and 12 ingestion of taboo substances) scenarios [9]. However, this work also needs close, independent replication.

# 6. Conclusion

This carefully designed and conducted replication study does not provide final or absolute evidence for the mental state × domain interaction effect, but it does add to the evidence for the effect. We think this (successful) replication moves us closer to understanding the mechanisms underlying moral cognition. Specifically, having a replicable phenomenon to explain is a necessary prerequisite for the challenging task of understanding the mechanisms that explain such regularities in our moral cognition. We suggest that the mental state × domain interaction effect offers researchers a potential foothold to begin such explanatory work.

Research ethics statement. Before participation in the online study, participants were given information about the general nature of the study without giving details of the research questions. Participants were informed that participation was voluntarily and that they could exit the study at any time without this affecting their payment. They were given details of how their data would be handled and contact details if they had concerns. The information sheet and informed consent is available in the OSF repository as part of the Materials. The materials used in the study had previously be granted ethical approval from the Psychology ethics committee (eCLESPsy001180 v. 3.4).

Data accessibility. All study data, scripts and stimulus material are available at an Open Science Framework (OSF) repository at https://osf.io/t7mzb/

Authors' contributions. J.S. and G.A.N.: (1) substantial contributions to conception and design, or acquisition of data, or analysis and interpretation of data; (2) drafting the article or revising it critically for important intellectual content; (3) final approval of the version to be published; and (4) agreement to be accountable for all aspects of the work in ensuring that questions related to the accuracy or integrity of any part of the work are appropriately investigated and resolved.

Competing interests. We declare we have no competing interests.

Funding statement. No funding has been received for this study.

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
