## [Reviewer comments · Royal Society Open Science]

Review History

RSOS-190808.R0 (Original submission)

Review form: Reviewer 1

Do you have any ethical concerns with this paper?

No

Have you any concerns about statistical analyses in this paper?

No

Recommendation?

Accept in principle

Comments to the Author(s)

This is well set up and clearly described. The authors have identified a study worth replicated, and explained its theoretical important concisely.

Review form: Reviewer 2 (Liane Young)

Do you have any ethical concerns with this paper?

No

Have you any concerns about statistical analyses in this paper?

No

Recommendation?

Accept in principle

Comments to the Author(s)

This submission looks good to me and aims to replicate prior work showing an intent x domain interaction such that intent matters more for moral judgments of harms compared to moral judgments of purity violations.

I have just one suggestion, to be taken with a grain of salt, as I am new to reviewing preregistered reports.

I think it would be useful to situate the proposed work in the complete context of work that has replicated the target effect (and related work in the literature).

For example, here are two other papers (fMRI, behavioral) that have revealed the domain x intent interaction:

Chakroff, A., Dungan, J., Koster-Hale, J., Brown, A., Saxe, R., & Young, L. (2016). When minds matter for moral judgment: intent information is neurally encoded for harmful but not impure acts. *Social Cognitive and Affective Neuroscience*, 11(3), 476–484. doi:10.1093/scan/nsv131

Chakroff, A., & Young, L. (2015). Harmful situations, impure people: An attribution asymmetry across moral domains. *Cognition*, 136(C), 30–37. doi:10.1016/j.cognition.2014.11.034

Here's what I think is the most compelling evidence for the domain x intent interaction, from a recent cross-cultural paper (in a nutshell, in 9 out of the 10 societies tested, and in all societies showing an effect of intent, they find the domain x intent interaction):

Barrett, H. C., Bolyanatz, A., Crittenden, A. N., Fessler, D. M. T., Fitzpatrick, S., Gurven, M., et al. (2016). Small-scale societies exhibit fundamental variation in the role of intentions in moral judgment. *Proceedings of the National Academy of Sciences*, 113(17), 4688–4693. doi:10.1073/pnas.1522070113

Here is related work showing that intent matters more for anger (linked to harm) than purity (linked to disgust):

Giner-Sorolla, R., Bosson, J. K., Caswell, T. A., & Hettinger, V. E. (2012). Emotions in sexual morality: Testing the separate elicitors of anger and disgust. *Cognition & Emotion*, 26(7), 1208–1222. doi:10.1080/02699931.2011.645278

Russell, P. S., & Giner-Sorolla, R. (2011). Moral anger is more flexible than moral disgust. *Social Psychological and Personality Science*, 2(4), 360–364. doi:10.1177/1948550610391678

Russell, P. S., & Giner-Sorolla, R. (2013). Bodily moral disgust: What it is, how it is different from anger, and why it is an unreasoned emotion. *Psychological Bulletin*, 139(2), 328–351. doi:10.1037/a0029319

Here is other work showing related differences between harm and purity:

Sabo, J. S., & Giner-Sorolla, R. (2017). Imagining wrong: Fictitious contexts mitigate condemnation of harm more than impurity. *Journal of Experimental Psychology: General*, 146(1), 134–153. doi:10.1037/xge0000251

Chakroff, A., Russell, P.S., Piazza, J., & Young, L. (2017). From impure to harmful: asymmetric expectations about immoral agents. *Journal of Experimental Social Psychology*, 69, 201–209.

Dungan, J.A., Chakroff, A., & Young, L. (2017) The relevance of moral norms in distinct relational contexts: Purity versus harm norms regulate self-directed actions. *PLoS ONE* 12(3): e0173405. doi:10.1371/journal.pone.0173405

Rottman, J. & Young, L. (in press). Specks of Dirt and Tons of Pain: Dosage Distinguishes Impurity from Harm. *Psychological Science*.

I also had one question: the authors said something about how in the original article all but one experiment (the one being replicated) displayed the manipulation in bold font. Can that be fact-checked?

And finally here are two small typos:

MFT claims that these modules each processes information – should be either “these modules process” or “each module processes”

“Young as Saxe” should be “Young and Saxe”

Decision letter (RSOS-190808.R0)

31-May-2019

Dear Dr Sweetman

On behalf of the Editors, I am pleased to inform you that your Manuscript RSOS-190808 entitled "Replicating different roles of intent across moral domains" deemed suitable for in-principle acceptance in Royal Society Open Science subject to minor revision in accordance with the referee and editor suggestions. Please find their comments at the end of this email.

The reviewers have recommended publication, but also suggest some minor revisions to your manuscript. Therefore, I invite you to respond to the comments and revise your manuscript.

Please you submit the revised version of your manuscript within 7 days (i.e. by the 08-Jun-2019). If you do not think you will be able to meet this date please let me know immediately.

When submitting your revised manuscript, you will be able to respond to the comments made by the referees and upload a file "Response to Referees" in the "File Upload" step. You can use this to document any changes you make to the original manuscript. In order to expedite the

processing of the revised manuscript, please be as specific as possible in your response to the referees.

Full author guidelines can be found here

<http://rsos.royalsocietypublishing.org/page/replication-studies#AuthorsGuidance>.

Kind regards,

Professor Chris Chambers

Registered Reports Editor

Associate Editor Comments to Author (Professor Chris Chambers):

Two expert reviewers have now assessed the Stage 1 submission. Both reviews are positive, with Reviewer 1 completely satisfied, and Reviewer 2 requesting minor revisions to the Introduction. Once these issues are addressed, Stage 1 acceptance should be forthcoming without requiring further in-depth review.

Reviewers' comments to Author:

Reviewer: 1

Comments to the Author(s)

This is well set up and clearly described. The authors have identified a study worth replicated, and explained its theoretical important concisely.

Reviewer: 2

Comments to the Author(s)

This submission looks good to me and aims to replicate prior work showing an intent x domain interaction such that intent matters more for moral judgments of harms compared to moral judgments of purity violations.

I have just one suggestion, to be taken with a grain of salt, as I am new to reviewing preregistered reports.

I think it would be useful to situate the proposed work in the complete context of work that has replicated the target effect (and related work in the literature).

For example, here are two other papers (fMRI, behavioral) that have revealed the domain x intent interaction:

Chakroff, A., Dungan, J., Koster-Hale, J., Brown, A., Saxe, R., & Young, L. (2016). When minds matter for moral judgment: intent information is neurally encoded for harmful but not impure acts. *Social Cognitive and Affective Neuroscience*, 11(3), 476–484. doi:10.1093/scan/nsv131

Chakroff, A., & Young, L. (2015). Harmful situations, impure people: An attribution asymmetry across moral domains. *Cognition*, 136(C), 30–37. doi:10.1016/j.cognition.2014.11.034

Here's what I think is the most compelling evidence for the domain x intent interaction, from a

recent cross-cultural paper (in a nutshell, in 9 out of the 10 societies tested, and in all societies showing an effect of intent, they find the domain x intent interaction):

Barrett, H. C., Bolyanatz, A., Crittenden, A. N., Fessler, D. M. T., Fitzpatrick, S., Gurven, M., et al. (2016). Small-scale societies exhibit fundamental variation in the role of intentions in moral judgment. *Proceedings of the National Academy of Sciences*, 113(17), 4688–4693. doi:10.1073/pnas.1522070113

Here is related work showing that intent matters more for anger (linked to harm) than purity (linked to disgust):

Giner-Sorolla, R., Bosson, J. K., Caswell, T. A., & Hettinger, V. E. (2012). Emotions in sexual morality: Testing the separate elicitors of anger and disgust. *Cognition & Emotion*, 26(7), 1208–1222. doi:10.1080/02699931.2011.645278

Russell, P. S., & Giner-Sorolla, R. (2011). Moral anger is more flexible than moral disgust. *Social Psychological and Personality Science*, 2(4), 360–364. doi:10.1177/1948550610391678

Russell, P. S., & Giner-Sorolla, R. (2013). Bodily moral disgust: What it is, how it is different from anger, and why it is an unreasoned emotion. *Psychological Bulletin*, 139(2), 328–351. doi:10.1037/a0029319

Here is other work showing related differences between harm and purity:

Sabo, J. S., & Giner-Sorolla, R. (2017). Imagining wrong: Fictitious contexts mitigate condemnation of harm more than impurity. *Journal of Experimental Psychology: General*, 146(1), 134–153. doi:10.1037/xge0000251

Chakroff, A., Russell, P.S., Piazza, J., & Young, L. (2017). From impure to harmful: asymmetric expectations about immoral agents. *Journal of Experimental Social Psychology*, 69, 201–209.

Dungan, J.A., Chakroff, A., & Young, L. (2017) The relevance of moral norms in distinct relational contexts: Purity versus harm norms regulate self-directed actions. *PLoS ONE* 12(3): e0173405. doi:10.1371/journal.pone.0173405

Rottman, J. & Young, L. (in press). Specks of Dirt and Tons of Pain: Dosage Distinguishes Impurity from Harm. *Psychological Science*.

I also had one question: the authors said something about how in the original article all but one experiment (the one being replicated) displayed the manipulation in bold font. Can that be fact-checked?

And finally here are two small typos:

MFT claims that these modules each processes information – should be either “these modules process” or “each module processes”

“Young as Saxe” should be “Young and Saxe”

Author's Response to Decision Letter for (RSOS-190808.R0)

See Appendix A.

Decision letter (RSOS-190808.R1)

06-Jun-2019

Dear Dr Sweetman

On behalf of the Editor, I am pleased to inform you that your Stage 1 Replication RSOS-190808.R1 entitled "Replicating different roles of intent across moral domains" has been accepted in principle for publication in Royal Society Open Science.

You may now progress to Stage 2 and complete the study as approved.

Please note that you must now register your approved protocol on the Open Science Framework (<https://osf.io/rr>), using the 'Submit your approved Registered Report' option and then the 'Registered Report Protocol Preregistration' option. Please use the Registered Report option even though your article is being accepted as a Stage 1 Replication. Please note that a time-stamped, independent registration of the protocol is mandatory under journal policy, and manuscripts that do not conform to this requirement cannot be considered at Stage 2. The protocol should be registered unchanged from its current approved state. Please include a URL to the protocol in your Stage 2 manuscript.

We would be grateful if you could now update the journal office as to the anticipated completion date of your study.

Following completion of your study, we invite you to resubmit your paper for peer review as a Stage 2 Replication. Please note that your manuscript can still be rejected for publication at Stage 2 if the Editors consider any of the following conditions to be met:

- The Introduction and methods deviated from the approved Stage 1 submission (required).
- The authors' conclusions were not considered justified given the data.

We encourage you to read the complete guidelines for authors concerning Stage 2 submissions at: <http://rsos.royalsocietypublishing.org/page/replication-studies#AuthorsGuidance>. Please especially note the requirements for data sharing and that withdrawing your manuscript will result in publication of a Withdrawn Registration.

Once again, thank you for submitting your manuscript to Royal Society Open Science and I look forward to receiving your Stage 2 submission. If you have any questions at all, please do not hesitate to get in touch. We look forward to hearing from you shortly with the anticipated submission date for your stage two manuscript.

Kind regards,
Professor Chris Chambers
Royal Society Open Science
openscience@royalsociety.org

Author's Response to Decision Letter for (RSOS-190808.R1)

See Appendix B.

RSOS-190808.R2 (Revision)

Review form: Reviewer 1

Is the manuscript scientifically sound in its present form?

Yes

Is the language acceptable?

Yes

Do you have any ethical concerns with this paper?

No

Have you any concerns about statistical analyses in this paper?

No

Recommendation?

Accept as is

Comments to the Author(s)

This looks great; well done. No notes.

Decision letter (RSOS-190808.R2)

01-May-2020

Dear Dr Sweetman:

It is a pleasure to accept your manuscript entitled "Replicating different roles of intent across moral domains" in its current form for publication in Royal Society Open Science. The comments of the reviewer(s) who reviewed your manuscript are included at the foot of this letter.

Kind regards,
Andrew Dunn
Royal Society Open Science
openscience@royalsociety.org

on behalf of Professor Chris Chambers (Associate Editor) and Chris Chambers (Registered Reports Editor, Royal Society Open Science)
openscience@royalsociety.org

Associate Editor Comments to Author (Professor Chris Chambers):

The Stage 2 manuscript was returned to one of the original Stage 1 reviewers, and I also read it closely myself. The reviewer is fully satisfied that the Stage 2 criteria were met, as am I, so we are in the happy position of being able to accept the manuscript as-is. My compliments to the authors.

There is one minor revision that the authors can make at the copyediting stage and which need not delay Stage 2 acceptance. On p3, lines 34-35, in the sentence "Following IPA, the accepted Stage 1 version of the manuscript was preregistered on the OSF at <https://osf.io/t7mzb/>", please replace the supplied URL (<https://osf.io/t7mzb/>) with the direct URL to the accepted Stage 1 manuscript: <https://osf.io/wjg63/> The link the authors have provided is to the general OSF project space rather than to the approved protocol specifically.

Reviewer(s)' Comments to Author:

Reviewer: 1

Comments to the Author(s)

This looks great; well done. No notes.

Appendix A

Dr J. Sweetman
Department of Psychology
College of Life and Environmental Sciences
University of Exeter
EX4 4QG
j.p.sweetman@exeter.ac.uk

Dear Professor Chambers,

Response to Referees (RSOS-190808)

My co-author (George Newman) and I are grateful for your decision to deem our manuscript “Replicating different roles of intent across moral domains” as suitable for in-principle acceptance in Royal Society Open Science, subject to minor revision. We would like to thank you and the reviewers for their constructive comments on our initial manuscript. We have now addressed the issues raised by Reviewer 2 and now believe that we have produced an improved manuscript as a result. Below we detail how we have responded to the issues raised.

Reviewer 1

This reviewer was completely satisfied.

Reviewer 2

This reviewer raised three issues:

Issue 1: “I think it would be useful to situate the proposed work in the complete context of work that has replicated the target effect (and related work in the literature).”

Response: We have now included additional details on the more recent converging evidence or “conceptual replications” of the mental state × domain interaction (3, p.2). Including all of the studies that reviewer 2 mentions would be outside of the scope of the present paper. Rather, we include the most pertinent examples: namely, Chakroff et al., (2016) and Barrett, et al., (2016).

Issue 2: “I also had one question: the authors said something about how in the original article all but one experiment (the one being replicated) displayed the manipulation in bold font. Can that be fact-checked?”

Response: After submitting the paper we heard back a second time from Liane Young who informed us that the mental state information was not presented in bold in any of the studies. Rather, the supplementary materials had this information in bold merely to highlight the experimental

manipulation to the reader. Accordingly, we have added this information to the method (4.1, p.2) and changed the materials to remove the bold type for the mental state information. This has also allowed us to delete the remarks concerning this as a possible difference between the original study and the replication (4.1.2, p. 3).

Issue 3: “And finally here are two small typos: MFT claims that these modules each processes information – should be either “these modules process” or “each module processes” “Young as Saxe” should be “Young and Saxe””

Response: We have corrected these typos and are grateful to the reviewer for spotting them.

Again, thank you for the time and consideration you have given this manuscript to date. We hope that the revisions meet with your approval. We look forward to hearing from you.

Best wishes,

Dr. Joseph Sweetman, Psychology, College of Life and Environmental Sciences, University of Exeter, Exeter, EX4 4QG, UK. j.p.sweetman@exeter.ac.uk

Co-authors' details:

Mr Newman, Psychology, College of Life and Environmental Sciences, University of Exeter, Exeter, EX4 4QG, UK. gn237@exeter.ac.uk

Appendix B

Dr J. Sweetman
Department of Psychology
College of Life and Environmental Sciences
University of Exeter
EX4 4QG
j.p.sweetman@exeter.ac.uk

Dear Professor Chambers,

Response to Referees (RSOS-190808)

My co-author (George Newman) and I are grateful for your decision to deem our manuscript “Replicating different roles of intent across moral domains” as suitable for in-principle acceptance in Royal Society Open Science. We now submit the Stage 2 document for review.

Thank you for the time and consideration you have given this manuscript to date. We hope that the Stage 2 document meets with your approval. We look forward to hearing from you.

Best wishes,

Dr. Joseph Sweetman, Psychology, College of Life and Environmental Sciences, University of Exeter, Exeter, EX4 4QG, UK. j.p.sweetman@exeter.ac.uk

Co-authors' details:

Mr Newman, Psychology, College of Life and Environmental Sciences, University of Exeter, Exeter, EX4 4QG, UK. gn237@exeter.ac.uk